# Correlations between Gut Microbiota and Hematological, Inflammatory, Biochemical and Oxidative Stress Parameters in Treatment-Naïve Psoriasis Patients

**DOI:** 10.3390/ijms25126649

**Published:** 2024-06-17

**Authors:** Elena Codruța Cozma, Ionela Avram, Vlad Mihai Voiculescu, Mara Mădălina Mihai, Amelia Maria Găman

**Affiliations:** 1Doctoral School, University of Medicine and Pharmacy of Craiova, 200349 Craiova, Romania; codrutadobrica@gmail.com; 2Department of Dermatology, “Elias” University Emergency Hospital, 011461 Bucharest, Romania; 3Department of Genetics, Faculty of Biology, University of Bucharest, 060101 Bucharest, Romania; ionela.sarbu@bio.unibuc.ro; 4Department of Oncologic Dermatology, “Elias” Emergency University Hospital, “Carol Davila” University of Medicine and Pharmacy, 020021 Bucharest, Romania; mara.mihai@umfcd.ro; 5Department of Pathophysiology, University of Medicine and Pharmacy of Craiova, 200349 Craiova, Romania; gamanamelia@yahoo.com; 6Clinic of Hematology, Filantropia City Hospital, 200143 Craiova, Romania

**Keywords:** microbiota, psoriasis, oxidative stress, hematological parameters, inflammatory parameters

## Abstract

Psoriasis is an inflammatory dermatosis with a complex pathogenesis, significantly impacting the quality of life of patients. The role of oxidative stress and gut microbiota in the pathogenesis of this disease is increasingly studied, appearing to underlie the comorbidities associated with this condition. We present the first prospective observational study conducted in Romania evaluating the interrelationship between gut microbiota and hematological, inflammatory, biochemical, and oxidative stress parameters in treatment-naïve psoriasis patients. Significant differences were observed in terms of microbiota composition, with lower levels of *Firmicutes* and *Enterobacteriaceae* in the psoriasis group compared to the control group. Moreover, a negative correlation was found between the serum triglyceride levels in patients with psoriasis and the *Enterobacteriaceae* family (*p* = 0.018, r = −0.722), and a positive correlation was found between the serum glucose levels and the *Firmicutes/Bacteroides* ratio (*p* = 0.03, r = 0.682). Regarding the oxidant–antioxidant status, a significant correlation was found between the FORT level and *Lactobacillus* (*p* = 0.034, r = 0.669). Lastly, the *Firmicutes* level negatively correlated with the DLQI level, independent of the clinical severity of the disease (*p* = 0.02, r = −0.685). In conclusion, even though the number of included patients is small, these results may serve as a starting point for future research into the involvement of the microbiota–inflammation–oxidative stress axis in psoriasis development.

## 1. Introduction

Psoriasis is one of the most common dermatological diseases, involving numerous genetic and environmental factors in its pathogenesis. The association of this condition with numerous systemic comorbidities, as well as the chronicity of cutaneous manifestations, results in a significant psycho-social impact on patients, leading to an impaired quality of life (QoL) [1].

Oxidative stress (OS) is a concept that is increasingly studied, with more and more studies revealing its involvement in the pathophysiology of multiple chronic diseases. In most specialized works, OS is defined as an imbalance generated by a transient or chronic increase in the level of reactive oxygen species (ROS) and/or a decrease in the activity of antioxidant systems. Initially, OS was defined exclusively as a phenomenon generated by a disease through its toxic cellular effects. Nowadays, special attention is also given to its intervention in the modulation of cellular functions [2,3,4].

The intestinal microbiota encompasses all microorganisms (bacteria, fungi, parasites, viruses, and archaea) that colonize the gastrointestinal tract, participating in a complex and dynamic relationship with the innate and adaptive immune system. Imbalances in the relationship between these microorganisms and the immune system, as well as the impact of external factors on microbiota composition, seem to be associated with the development of chronic diseases such as inflammatory bowel diseases, celiac disease, psoriasis, atopic dermatitis, neurodegenerative diseases, and cardiovascular diseases [5,6]. Understanding these mechanisms and how these microorganisms interact with the immune system may represent an important step toward a more complex understanding of the mechanisms of these diseases.

Regarding the relationship between the OS level and the development of psoriasis, recent studies highlight significant changes in oxidant–antioxidant balance in these patients. At the same time, more and more studies are trying to decipher the role that the intestinal microbiota has in the development and progression of psoriasis, thus finding the presence of an intestinal dysbiosis in all patients with psoriasis, with a quantitative change in the main phyla at the intestinal level [7,8].

However, the interrelationship between microbiota and OS in these patients has not yet been evaluated, although the hypothesis of increased OS as a result of the dysbiosis process has been presented. Moreover, some therapies already used in psoriasis with significant efficiency are known not only for their effect of decreasing OS but also for modulating the composition of the skin microbiota.

The purpose of this study is to evaluate the relationship between the oxidative status of the organism (assessed through FORT and FORD tests) and the composition of the intestinal microbiota (assessed through the quantitative polymerase chain reaction (qPCR) method), as well as the hematological and inflammatory parameters in a group of treatment-naïve patients diagnosed with moderate to severe psoriasis. Furthermore, within the same study, the presence or absence of significant alterations in the composition of the microbiota in naïve psoriasis patients versus a control group will be evaluated.

## 2. Results

The study group consisted of 10 patients with moderate to severe psoriasis, who were treatment-naïve and who met the inclusion and exclusion criteria shown in Figure 1. The control group comprised 10 healthy volunteers matched for age, sex, and place of origin, who met the previously stated exclusion criteria. The study group had a mean age of 47.9 years (+/−12.59), with 50% females, while the control group had a mean age of 40 years (+/−14.73), with 25% females. The main characteristics, as well as the mean values of the evaluated parameters (hematological, inflammatory, biochemical, oxidative stress, total antioxidant capacity, disease severity, and its impact on quality of life), are presented in Table 1. 

Regarding the results obtained through qPCR, a significant difference was observed between the bacteria of the Firmicutes phylum in the study group compared to the control group, with a decrease in the former (*p* = 0.03). Additionally, a significant decrease in microorganisms from the Enterobacteriaceae family was also noted in the psoriasis group compared to the control group (*p* < 0.0001). No significant differences were observed between the two groups for the *Lactobacillus* group, *Bacteroides* group, *Bifidobacterium* genus, *Actinobacteria*, and *Prokaryotes* (Figure 1).

Regarding the relationship between various bacterial families/groups and the levels of inflammatory and hematological parameters, the results are presented in Table 2 and Table 3, respectively. Principal component analysis was performed for the inflammatory parameters (MLR, PLR, LMR, NLR, SII, MPV/P, PCR, ESR, and fibrinogen) and the evaluated microorganisms and the loadings plot are presented in Figure 2.

Analyzing metabolic parameters, a negative correlation was found between the serum triglyceride levels in patients with psoriasis and the level of microorganisms from the *Enterobacteriaceae* family (*p* = 0.018, r = −0.722), and a positive correlation was found between the serum glucose levels and the *Firmicutes/Bacteroides* ratio (*p* = 0.03, r = 0.682).

Regarding the oxidant and antioxidant status, a significant correlation was found between the FORT level and *Lactobacillus* (*p* = 0.034, r = 0.669). No statistically significant correlations were observed between the levels of other groups of microorganisms and the FORT or FORD levels.

Lastly, the *Firmicutes* level negatively correlated with the DLQI level, independent of the clinical severity of the disease (*p* = 0.02, r = −0.685).

## 3. Materials and Methods

### 3.1. Study Group 

The study was conducted between September 2023 and March 2024 and included patients with moderate-to-severe psoriasis who met the inclusion and exclusion criteria mentioned below.

Within the study, patients were included using a non-probability sampling–judgmental (purposive) sampling method. The study group comprised 10 patients with moderate-to-severe psoriasis, originating from urban areas and with no history of systemic treatment, who met the inclusion and exclusion criteria. The control group involved the selection of 10 healthy volunteers who were matched in terms of age, sex, and environmental background with the study group. The inclusion and exclusion criteria for selecting patients, as well as the healthy volunteers included in the study, are outlined below (Table 4).

### 3.2. Ethical Aspects

The present study was conducted with the approval of the Ethics Committee of the University of Medicine and Pharmacy in Craiova (approval no. 42/2 April 2021) and the Emergency University Hospital Bucharest (approval no. 1092/17 February 2023), Romania. The inclusion of patients in the study, data collection, and biological sample collection involved a preliminary stage of informing them about the study procedure and obtaining their signed consent forms. Patients’ rights were respected according to the guidelines of the World Health Organization and the Helsinki Declaration.

### 3.3. Protocol

For each patient included in the study, we evaluated their hematological, inflammatory, biochemical, oxidative stress, disease severity parameters, and gut microbiota composition. The parameters evaluated for each patient are represented in Figure 3. 

#### 3.3.1. Assessing Oxidant Status through Free Oxygen Radical Test (FORT) Evaluation

The assessment of oxidant status was performed using the FORT AD-12107-A kit from Callegari 1930, Parma, Italy. The determination of FORT is carried out from capillary blood, after collecting 20 μL of capillary blood in the capillary tube provided with the kit and introducing the collected sample into the tube with reagent R2 (containing a buffer solution with an acidic pH). The contents of the resulting sample are poured into the cuvette of reagent R1 and centrifuged at 3000 rpm for 1 min. The last step involves reading the sample for 6 min in the CR3000 Analyzer, preheated to 37 °C. The result is expressed in FORT units (1 FORT unit is equivalent to 0.26 mg/L H_2_O_2_), with normal values considered to be below 310 FORT units (2.3 mmol/L H_2_O_2_). The test is based on the Fenton reaction, which involves the release of iron ions and the production of alkoxyl and peroxyl radicals. Upon the radicals coming into contact with diphenyl-diamine derivatives, this leads to the production of a colored compound that can be measured spectrophotometrically.

#### 3.3.2. Assessing Antioxidant Status through FORD Evaluation

The assessment of antioxidant status was conducted using the FORD AD-12136 kit from Callegari 1930, Parma, Italy. Determining FORD status involves collecting 50μL of capillary blood into the capillary tube provided by the kit, introducing the collected sample into the tube with reagent S1, and centrifuging it for 1 min at 3000 rpm. Subsequently, the reagent is transferred from tube S2 to cuvette C1, followed by the addition of 50 μL of reagent S3. Then, the first reading (4 min) is performed by the preheated CR3000 analyzer at 37 °C. After completing the first reading, 100μL of supernatant from the centrifuged sample S1 is added. The final step involves the second reading (2 min). The results are expressed in Trolox units (1 Trolox unit is equivalent to 0.25–0.3 mmol/L), with normal values considered to be in the range of 1.07–1.53 Trolox units. The test is based on the ability of dimethylamine sulfate to form a colored compound, together with the iron ions released in the solution, which is measured spectrophotometrically. This compound will be reduced by the antioxidants in the blood, resulting in a loss of coloration of the solution when measured spectrophotometrically.

#### 3.3.3. Assessing Psoriasis Severity and the Impact on the Quality of Life of Psoriasis Patients

The assessment of disease severity was conducted by the study investigator through the completion of the psoriasis assessment severity index (PASI) form following the clinical examination of each patient. The PASI score can range from 0 to 72. The disease severity assessment was performed at the time of the patient’s inclusion in the study.

The assessment of the disease’s impact on the patient’s quality of life was conducted by each patient through the completion of the disease life quality index (DLQI) form. The DLQI score can range from 0 to 30. 

#### 3.3.4. Assessing Gut Microbiota Composition

The evaluation of intestinal microbiota composition in both the study and control groups involved the preliminary steps of stool sample collection, preservation, and DNA extraction.

Stool sample collection and preservation were performed in specialized collection containers containing sterile glycerol solution, followed by immediate freezing at −20 °C by the patient or a healthy volunteer until transportation to the laboratory. Samples were stored at a temperature of −70 °C until further processing at the Genetics Laboratory of the Faculty of Biology, University of Bucharest, Romania.

DNA extraction was conducted using the QiAamp DNA microbiome kit (Qiagen, Germantown, MD, USA); a stool sample was homogenized and 1 mL was used for DNA extraction. Following DNA extraction, the concentration and purity of the obtained sample were assessed using a NanoVue Plus spectrophotometer (GE, Boston, MA, USA).

Analysis by quantitative polymerase chain reaction (qPCR) was conducted using 10 ng of DNA with the GoTaq qPCR Master Mix kit (Promega, Madison, WI, USA). Measurements were performed in triplicate using the RT-PCR 7900 analyzer (Applied Biosystems, Foster City, CA, USA). Genome copies of 6 bacterial groups were assessed (study and control): the *Lactobacillus* group, *Firmicutes* phylum, *Bacteroides* group, *Bifidobacterium* genus, *Enterobacteriaceae* family, and *Actinobacteria* phylum. The steps of the qPCR process are schematically represented in Figure 4. The description of the primers used and the protocol for determining the number of copies (according to standard curves) have been described in previous publications [9,10]. 

#### 3.3.5. Assessing Hematological Parameters and Inflammatory Markers

In addition to the previously enumerated parameters, each patient underwent an evaluation of standard laboratory parameters and inflammatory markers (Figure 3). Venous blood samples were collected in the morning, after overnight fasting. Hematological parameters were measured using the SYSMEX_XN_HEMA analyzer, while other biochemical parameters and C-reactive protein (CRP) levels were assessed using the ARCHITECT c8000 analyzer. Fibrinogen levels were determined using the ACLTOP550H analyzer, and the erythrocyte sedimentation rate at 1 h (ESR1h) was measured using the SEDIPLUS 2000 device. The evaluation of biological parameters was conducted at the time of study inclusion. Within the study, the following inflammatory markers were calculated: MLR (monocytes/lymphocytes ratio), LMR (lymphocytes/monocytes ratio), PLR (platelets/lymphocytes ratio), NLR (neutrophils/lymphocytes ratio), SII (systemic inflammatory index), and MPV/PLT (mean platelet volume/platelets ratio). The calculation method and significance of these markers have been presented in a previous study [11].

### 3.4. Statistical Analysis

A one-way ANOVA followed by Tukey’s multiple comparisons test was performed to assess the differences between the study group and the control group for each measured bacterial group. Pearson correlations were conducted to evaluate the relationship of FORT, FORD, PASI, DLQI, hematological, biochemical, and inflammatory variables with each bacterial group. Principal component analysis (PCA) was performed for inflammatory markers and microorganisms. All tests were conducted using GraphPad Prism version 10.0.0 for Mac OS X (GraphPad Software, Boston, MA, USA). A *p*-value of < 0.05 was considered significant.

## 4. Discussions

The present study is the first prospective observational study conducted in Romania to evaluate the composition of the gut microbiota in naïve patients with moderate-to-severe psoriasis, compared to the control group. Additionally, it is the first study to assess the interrelationship between the different bacterial groups present in the intestinal tract of these patients and hematologic, inflammatory, and biochemical parameters, total antioxidant capacity, and oxidant status levels.

Although most studies in the current literature evaluating the level of microorganisms from the *Firmicutes* phylum in patients with various forms of psoriasis (vulgaris, pustular, or psoriatic arthritis) show an alteration compared to the level found in healthy volunteers, the results are contradictory, with both decreases and increases observed, depending on the study [12].

For instance, a study conducted by Dei-Cas et al., analyzing the composition of the intestinal microbiota in untreated patients with psoriasis (divided into severity groups) versus a control group of healthy volunteers, found significant quantitative changes between different bacterial groups. The evaluation method involved 16S rRNA sequencing. It was found that there was a change in the percentages of the main phyla compared to the control group, with a decrease in the percentages of *Bacteroidetes* (47.1% versus 59.9%) and an increase in *Firmicutes* (44.6% versus 33%) and *Proteobacteria* (5.4% versus 4.2%). Significant differences were observed in the *Firmicutes/Bacteroidetes* ratio (*p* = 0.0002), with an increase observed in patients with psoriasis. The study did not identify changes in alpha diversity [13].

Similar results, with an increase in the *Firmicutes* and *Actinobacteria* phyla, were reported by Hidalgo-Cantabrana and colleagues in a study evaluating microbiota composition through 16S rRNA sequencing in 19 patients with psoriasis (*p* < 0.001). Furthermore, they highlighted that although the main bacterial phyla are present in both healthy volunteers and patients with psoriasis, there is a significant alteration in the proportions between them, especially in the *Bacteroidetes/Firmicutes* ratio [14].

Additionally, another study conducted by Huang et al., which evaluated microbial composition in 35 patients with psoriasis, identified a significant decrease in the percentage of *Firmicutes* at the intestinal level compared to the control group (*p* = 0.026), with an increase in the *Bacteroidetes* phylum (*p* < 0.0001). However, *Firmicutes* remained the dominant phylum at the intestinal level [15]. A decrease in *Firmicutes* has also been observed in other inflammatory diseases, such as Crohn’s disease and ulcerative colitis, as well as in chronic conditions such as left ventricular hypertrophy, type 2 diabetes mellitus, or cardiovascular diseases [16,17,18,19]. Furthermore, a decrease in *Firmicutes* has also been observed in the elderly, with a decrease in the *Firmicutes/Bacteroidetes* ratio as age advances [18].

Within our cohort, a decrease in the *Firmicutes* phylum was observed in treatment-naïve patients with moderate to severe psoriasis compared to the healthy control group (*p* = 0.03), but without a significant alteration of *Bacteroides* or the *Firmicutes/Bacteroides* ratio. The change in *Firmicutes* level is consistent with results from other studies in the literature described above.

The role that bacteria from the *Firmicutes* phylum play in modulating the inflammatory response is reported in numerous studies, both in healthy individuals and in those suffering from chronic diseases. For instance, a study by Martinez et al., which evaluated how a fiber-rich diet influenced intestinal microbiota in 28 healthy individuals, highlighted a negative correlation between the presence of the *Ruminococcaceae* family from the *Firmicutes* phylum and CRP (r = −0.59, *p* = 0.0024). Additionally, within this study, two genera (*Ruminococcus* and *Faecalibacterium*) showed a statistically significant negative correlation with fibrinogen and the plasma level of CRP (r = −0.48, *p* < 0.05 and r = −0.60, *p* < 0.01, respectively) [20]. 

Moreover, the bacteria from the phylum *Firmicutes* seem to be responsible for an anti-inflammatory effect, especially due to their role in the synthesis of butyrate and other short-chain fatty acids (SCFAs). Samaddar et al. demonstrated, in a prospective interventional study carried out on an animal model, a decrease in the level of intestinal *Firmicutes* in animal models that was characterized by an increase in the level of serum inflammation. This is explained by the increase in the level of lipopolysaccharide-producing bacteria, which will determine the activation of phagocytic cells toward a pro-inflammatory phenotype [21,22]. In our study, we also identified a negative correlation between the *Firmicutes* phylum and the level of fibrinogen (*p* = 0.02, r = −0.701) and CRP (*p* = 0.005, r = −0.803), with these results further emphasizing the role of these microorganisms in modulating the anti-inflammatory response. Moreover, the increased levels of inflammatory markers in these patients can be explained by the decrease in expression of microorganisms from the *Firmicutes* phylum, and, implicitly, of some anti-inflammatory metabolites such as SCFAs.

Besides its role in modulating the inflammatory status, the intestinal microbiota plays an important role in establishing a complex interrelationship between the nervous, gastrointestinal, endocrine, and immune systems, with variations in its composition and diversity being associated with the presence of psychiatric disorders (depression, autism spectrum disorders, anxiety, etc.) [23]. Psychiatric disorders represent a significant comorbidity of psoriasis, with chronic inflammatory status exacerbating not only the evolution of dermatological disease but also the associated comorbidities. Thus, psoriasis is associated with anxiety disorders, eating disorders, mood disorders, schizophrenia, sexual dysfunction, sleep disorders, and substance dependence or abuse. These are based on immune cell and nervous system alterations, along with skin barrier changes, as well as factors related to how psoriasis influences the quality of life of these patients (itching, lack of social support, stigmatization, etc.) [24,25,26].

Moreover, multiple studies highlight the role that intestinal microbial diversity has in inducing psychiatric disorders, especially through decreased anti-inflammatory metabolites such as SCFAs (butyrate). Thus, a study by Huang et al., evaluating intestinal microbiota composition through 16S rRNA sequencing in 27 patients with major depressive disorders compared to a healthy control group, highlighted a decrease in diversity indices in these patients. Furthermore, the *Firmicutes* phylum was significantly reduced compared to the control group (*p* < 0.01) [27]. Similar results were reported by Jiang et al., who highlighted decreased levels of *Firmicutes*, as well as an increase in *Bacteroidetes*, *Proteobacteria*, and *Actinobacteria* [28].

Although there are currently no studies evaluating the relationship between intestinal microbiota and psychiatric status or DLQI in patients with psoriasis, within our study, we identified a negative correlation between *Firmicutes* level and DLQI (*p* = 0.02, r = −0.685), the latter being the most important score used to evaluate the impact of the disease on patients’ quality of life and, implicitly, on their psychiatric status. Also, in a study conducted by Ali et al., the authors highlight that the DLQI level is strongly correlated with depressive symptoms, which are assessed through the hospital anxiety and depression scale (HDS) score (r = 0.715) [29].

Our results are consistent with data from the literature, with low levels of *Firmicutes* being associated with a higher frequency of psychiatric disorders [27,28,30]. Thus, a high level of DLQI, correlated with a higher frequency of depression in these patients, may be associated with a decrease in the *Firmicutes* phylum and implicitly with the increased peripheral expression of pro-inflammatory molecules, with an increase in IL-6 and TNF-alpha [27].

Regarding microorganisms from the *Enterobacteriaceae* family, these are represented by Gram-negative facultatively anaerobic bacteria, which are expressed in small percentages in the intestinal tract but play an essential role in oxygen consumption and in creating an anaerobic environment for the main microorganisms colonizing the gastrointestinal tract. An increase in their number at the intestinal level is observed in patients with inflammatory bowel diseases, as well as in cases of intestinal dysbiosis due to other causes (including systemic antibiotic treatment). However, it has not been established whether the increase in these microorganisms is a cause or an effect of intestinal dysbiosis [22,31,32]. Currently, there are no studies in the literature evaluating the levels of microorganisms from the *Enterobacteriaceae* family in patients with moderate to severe psoriasis. However, several studies have reported differences regarding the *Proteobacteria* phylum, with contradictory results. For example, Gao et al. demonstrated, in a cross-sectional study conducted with 6 patients with psoriasis and 6 control patients, a decrease in the expression of microorganisms from this phylum, while Drago et al. highlighted an increase. In our study, we observed a decrease in the number of copies of microorganisms from the *Enterobacteriaceae* family compared to the control group (*p* < 0.0001), which is consistent with the results obtained by Gao et al. [33]. However, current studies are conducted on small patient cohorts, with contradictory results, and adopt different methods of evaluating microbiota composition, making it difficult to draw clear conclusions in this regard.

Microorganisms from the genus *Lactobacillus* represent one of the most important constituents of intestinal microbiota, with roles in anti-inflammatory, antioxidant, local and systemic immune modulation, antineoplastic, and maintenance of intestinal epithelial integrity and health activities. These roles are exerted through the pleomorphic effects of this bacterial genus, namely: modulation of gene expression (Muc 2 and 3 genes, which are essential in intestinal mucus synthesis), synthesis of antimicrobial products, metabolism of mycotoxins, modulation of the signaling pathways involved in reactive oxygen species production, and the synthesis of antioxidant enzymes [34,35]. Thus, concerning their role in redox status, they have a strong antioxidant action, not only through the presence of numerous genes resistant to OS (thioxin reductase and thioxin catalase, glutathione reductase, catalase, and DNA protection proteins) but also through its effect of stimulating the Nrf-2/Keap 1 pathway and inhibiting NFκB [35]. Psoriasis is characterized not only by an increase in oxidative stress levels but also by a decrease in antioxidant mechanisms, with multiple studies highlighting the presence of increased levels of OS markers [7]. Furthermore, in a previous study, we found significantly increased levels of OS as evaluated by FORT in a group of 53 patients with psoriasis, regardless of the type of treatment compared to the control group. Moreover, statistically significant differences were found between FORT levels in treatment-naïve patients compared to those receiving biological therapy (*p* = 0.033) or classical systemic therapy (*p* = 0.04) [11]. Although there are no studies evaluating the level of *Lactobacillus* in patients with psoriasis, based on disease progression and severity stages, it has been observed that in other inflammatory conditions such as inflammatory bowel diseases, lower levels of *Lactobacillus* are associated with higher severity (*p* < 0.001, r = −0.559) [36]. Furthermore, another study conducted by Gao et al. identified lower levels of *Lactobacillus* in patients with atherosclerosis, with *Lactobacillus* levels being an independent predictor of disease severity [37]. In this current study, we identified a significant correlation between FORT level and *Lactobacillus* (*p* = 0.034, r = 0.669) in patients with moderate to severe psoriasis who were treatment-naïve. Additionally, although not statistically significant, in our cohort, we also observed a decrease in the level of *Lactobacillus* in patients with psoriasis compared to the control group. This relationship between FORT and *Lactobacillus* levels can be explained by the relatively short duration of the disease in our psoriasis group, approximately 4.55 ± 3.43 years, with the possible presence of a compensatory mechanism aiming to increase antioxidant systems in order to decrease the level of reactive oxygen species.

Regarding the correlations with hematological parameters, we observed a positive correlation between the level of monocytes and *Bifidobacterium* (*p* = 0.01, r = 0.765) and *Actinobacterial* (*p* = 0.01, r = 0.703), as well as between the level of basophils and *Lactobacillus* (*p* = 0.023, r = 0.706). A negative correlation was observed between the level of *Bacteroides* and eosinophils (*p* = 0.05, r = −0.632). Although there are still no studies in the literature that evaluate these interrelationships, and their interpretation in a clinical context is difficult, an explanation of this interrelationship can be suggested by the modulation of these cells (especially monocytes) by molecules with pleiotropic action secreted by these microorganism (SCAFs) [22].

Lastly, the correlations between inflammatory indices (NLR, MLR, SII, and PLR) and microorganisms from the *Bacteroides* and *Actinobacteria* genera are consistent with the roles of these microorganisms in the development of inflammatory diseases.

In a study conducted by Zang et al., the authors evaluated how the intestinal microbiota influences the development of psoriasis and/or psoriatic arthritis, with the *Bacteroidetes* phylum and the *Prevotella* genus being associated with a protective mechanism (*p* = 0.033 and *p* = 0.045, respectively) [38]. Furthermore, He et al. emphasized the protective role of *Bacteroides* in TNF-alpha-induced inflammation, the latter being a key cytokine involved in the pathophysiology of psoriasis [39].

Moreover, bacteria from the *Actinobacteria* phylum play an important role in the metabolism of SCFAs, being responsible for producing significant amounts of these metabolites, especially butyrate and its derivatives, despite their low representation among intestinal bacteria. The latter has pleomorphic effects, being important in maintaining intestinal barrier integrity (by increasing the expression of MUC2 genes and consequently intestinal mucus synthesis) and modulating inflammatory responses (by inducing prostaglandin synthesis), as well as modulating gene transcription (through epigenetic effects) [40]. 

Thus, the inverse association found in our study between *Bacteroides* and PLR (*p* = 0.03, r = −0.694), NLR (*p* = 0.02, r = −0.709), and MLR (*p* = 0.02, r = −0.697), as well as between *Actinobacteria* and LMR (*p* = 0.04, r = −0.655), once again emphasizes how a depletion in these microorganisms (with their anti-inflammatory and immunomodulatory roles) is correlated with an increase in those indices (which are predictors of disease severity, of acute and chronic inflammation, and immune system dysregulation) [40]. 

## 5. Study Limitations

One of the main limitations of this study is represented by the size of the study cohort and the control group. However, identifying and selecting patients with moderate to severe psoriasis who were treatment-naïve and who met the aforementioned inclusion and exclusion criteria was extremely challenging. The frequent use of antibiotics in Romania also led to the exclusion of a significant percentage of identified patients, together with the presence of comorbidities or unhealthy behaviors that could increase oxidative stress. Additionally, selecting both the control and study cohorts required careful consideration when choosing patients with similar diets who were from the same urban environment, as all these variables could significantly influence the composition of the microbiota. Therefore, to confirm the accuracy of statistically significant results in this cohort, further studies with larger patient cohorts are necessary, including comparative studies based on treatment type.

Another limitation of the study is the inability to visualize dynamic changes in intestinal microbiota following psoriasis treatment and improvement by way of PASI and DLQI scores, which will be analyzed in future studies.

Last but not least, conducting the FORT and FORD tests described earlier requires immediate processing of the samples in order to obtain accurate results, due to the rapid degradation of reactive oxygen species. Although sample processing was conducted within the first 30 min of collection for all patients in the study and the control groups, variations in results may still exist.

## 6. Conclusions

In conclusion, this is the first study to evaluate the relationship between microbiota composition, hematological, biochemical, inflammatory, and oxidative stress parameters, and disease severity scores (PASI and DLQI) in treatment-naïve patients with moderate to severe psoriasis. Within this study, we observed not only changes in the microbiota composition of psoriasis patients compared to the control group, with a decrease in microorganisms from the *Firmicutes* phylum and *Enterobacteriaceae* family, but also correlations between inflammatory parameters (CRP and fibrinogen) and *Firmicutes*, between FORT level and *Lactobacillus* genus, as well as between different metabolic parameters (triglycerides and serum glucose) and *Enterobacteriaceae* level, along with the *Firmicutes/Bacteroides* ratio. Although, currently, multiple studies in the literature evaluate separately the impact of microbiota on psoriasis development and the alteration of oxidative status, this is the first study that evaluates the interrelation between these parameters. Also, this is the first study that evaluates the correlation between inflammatory parameters and gut microbiota composition, underlining once again the involvement of microorganisms in chronic inflammation. Last but not least, the correlation between DLQI score and *Firmicutes* underscores, once again, the skin–gut axis relationship. Therefore, based on the promising observations made, this study will serve as a starting point for future research with larger patient cohorts, utilizing intestinal microbiota sequencing methods and monitoring patients before and after the initiation of psoriasis therapy over time.

## Figures and Tables

**Figure 1 ijms-25-06649-f001:**
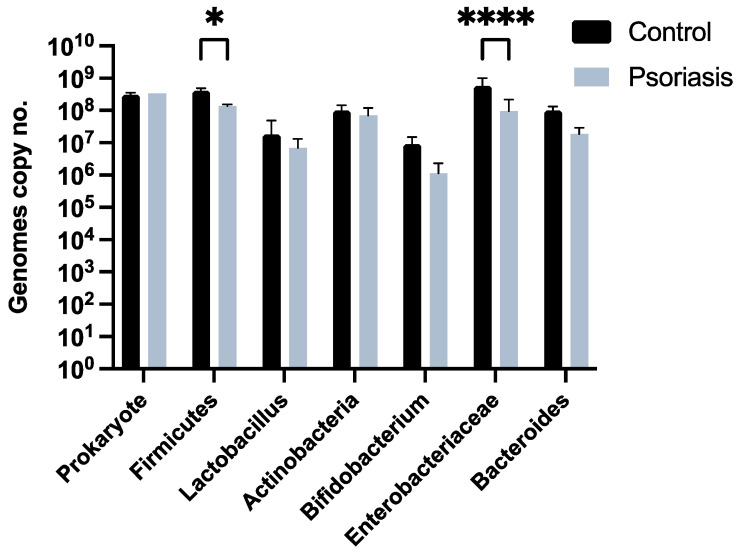
The number of genome copies based on the control group and the study group for the evaluated microorganisms. * *p* < 0.05, **** *p* < 0.0001.

**Figure 2 ijms-25-06649-f002:**
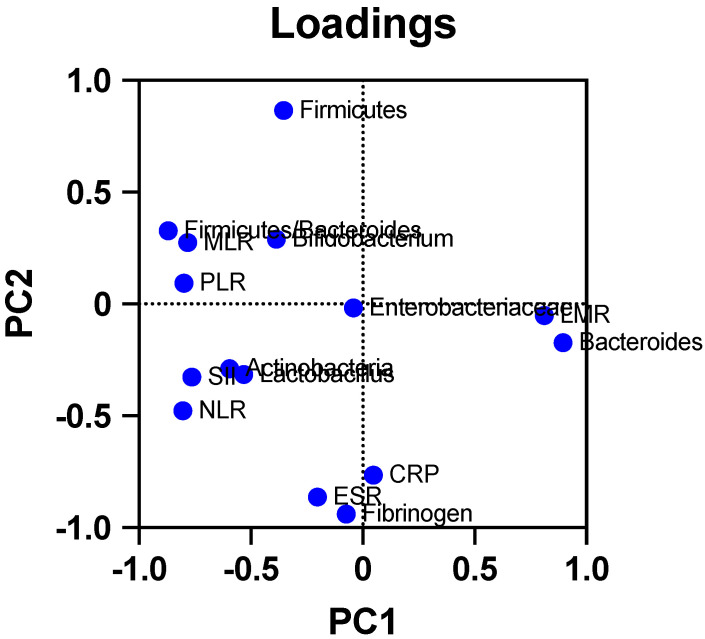
Loading plot from PCA for inflammatory parameters and evaluated microorganisms.

**Figure 3 ijms-25-06649-f003:**
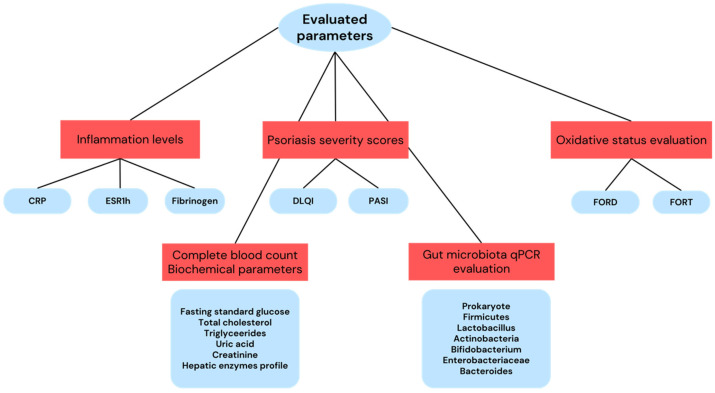
Evaluated parameters for the patients included in the study. CRP—C-reactive protein, ESR1h—erythrocyte sedimentation rate at 1 h, DLQI—disease life quality index, PASI—psoriasis area severity index, FORD—free oxygen radical defense, FORT—free oxygen radical test. Figure created by E.C. Dobrica (Cozma) in Canva Pro.

**Figure 4 ijms-25-06649-f004:**
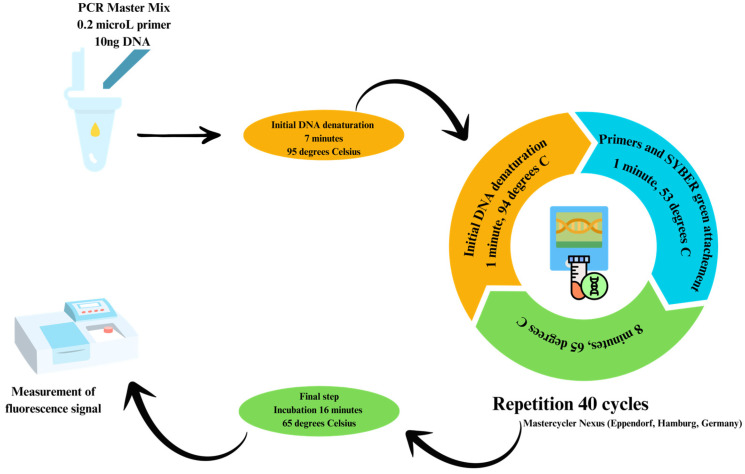
The steps required for conducting qPCR on stool samples in both the control and study groups. Figure created by E.C. Dobrica (Cozma) in Canva Pro.

**Table 1 ijms-25-06649-t001:** The characteristics of the study group—demographic, hematological, inflammatory, biochemical, oxidative stress, and disease severity parameters.

Parameter	Mean Value ± SD ^20^	Unit	Normal Range
Sex	50%	% Female	-
Age	47.9 ± 12.59	Years	-
Disease duration time	4.55 ± 3.43	Years	-
**Hematological parameters**			
WBC ^1^	7.23 ± 2.31	×10^3^/μL	4.6–10.2
Neutrophils	4.52 ± 2.03	×10^3^/μL	1.5–6.9
Lymphocytes	1.95 ± 0.56	×10^3^/μL	0.6–3.4
Monocytes	0.53 ± 0.15	×10^3^/μL	0.0–0.9
Eosinophils	0.19 ± 0.11	×10^3^/μL	0.0–0.7
Basophils	0.03 ± 0.01	×10^3^/μL	0.0–0.2
RBC ^2^	5.02 ± 0.44	×10^3^/μL	3.8–5.3
Hb ^3^	14.59 ± 1.23	g/dL	11.7–15.9
HTC ^4^	44.02 ± 1.23	%	35.0–47.0
Platelets	297.9 ± 106.5	×10^3^/μL	150–400
**Inflammation parameters**			
ESR ^5^	15 ± 12.16	mm/1 h	0–15
CRP ^6^	7.23 ± 6.46	mg/dL	<10
Fibrinogen	362.7 ± 145.3	mg/dL	238–498
MLR ^7^	0.29 ± 0.14	-	-
LMR ^8^	3.93 ± 1.4	-	-
PLR ^9^	160.7 ± 56.28	-	-
NLR ^10^	2.50 ± 1.25	-	-
SII ^11^	760.9 ± 448.6	-	-
MPV/PLT ^12^	0.039 ± 0.013	-	-
**Biochemical parameters**			
Glucose	96.6 ± 27.3	mg/dL	70–115
Urea	28.9 ± 11.61	mg/dL	19–43
Uric Acid	5.65 ± 2.15	mg/dL	2.0–6.2
Creatinine	0.78 ± 0.14	mg/dL	0.5–1.2
Triglycerides	109 ± 61.53	mg/dL	35–150
Total Cholesterol	210 ± 44.85	mg/dL	140–200
ALT ^13^	29.5 ± 16.6	U/L	6–55
AST ^14^	23.9 ± 9.27	U/L	11–34
GGT ^15^	30.5 ± 16.43	U/L	0–30
Total Bilirubin	0.65 ± 0.21	mg/dL	0.2–1.2
**Oxidative status parameters**			
FORT ^16^	273.3 ± 132	FORT units	<310
FORD ^17^	0.34 ± 0.14	Trolox units	1.07–1.53
**Disease severity parameters**			
PASI ^18^	12.65 ± 8.37	points	0
DLQI ^19^	12.9 ± 2.33	points	0

^1^ WBC—white blood cells, ^2^ RBC—red blood cells, ^3^ Hb—hemoglobin, ^4^ HTC—hematocrit, ^5^ ESR—erythrocytes sedimentation rate, ^6^ CRP—C-reactive protein, ^7^ MLR—monocytes/lymphocytes ratio, ^8^ LMR—lymphocytes/monocytes ratio, ^9^ PLR—platelets/lymphocytes ratio, ^10^ NLR—neutrophils/lymphocytes ratio, ^11^ SII—systemic inflammatory index, ^12^ MPV/PLT—mean platelet volume/platelets ratio, ^13^ ALT—alanine transaminase, ^14^ AST—aspartate transaminase, ^15^ GGT—gamma-glutamyl transferase, ^16^ FORT—free oxygen radical test, ^17^ FORD—free oxygen radical defense, ^18^ PASI—psoriasis assessment severity index, ^19^ DLQI—dermatology life quality index, ^20^ SD—standard deviation.

**Table 2 ijms-25-06649-t002:** Correlations between microorganisms and inflammatory parameters in psoriasis patients.

Microorganism	Inflammatory Markers	*p*-Value	Pearson r Coefficient
*Actinobacteria*	LMR	0.04	−0.655
*Bacteroides*	SII	0.05	−0.617
	MLR	0.02	−0.697
	PLR	0.03	−0.694
	NLR	0.02	−0.709
*Firmicutes*	Fibrinogen	0.02	−0.701
	CRP	0.005	−0.803
*Firmicutes*/*Bacteroides*	MLR	0.01	0.767
	PLR	0.02	0.680

**Table 3 ijms-25-06649-t003:** Correlations between microorganism and hematological parameters in psoriasis patients.

Microorganism	Hematological Parameter	*p*-Value	Pearson r Coefficient
*Actinobacteria*	Monocytes	0.01	0.730
*Bacteroides*	Eosinophils	0.05	−0.632
*Lactobacillus*	Basophils	0.02	0.706
*Bifidobacterium*	Monocytes	0.01	0.765

**Table 4 ijms-25-06649-t004:** Inclusion and exclusion criteria for patient selection.

Inclusion Criteria	Exclusion Criteria
Histopathological confirmation of psoriasis diagnosis	Chronic alcohol consumption and smoking
Moderate to severe psoriasis: - DLQI > 10; - 1 of 3 criteria (PASI ≥ 11 or BSA ≥ 10 or sPGA ≥ 3) and DLQI ≥ 5; or - 2 of 3 criteria (PASI ≥ 11 or BSA ≥ 10 or sPGA ≥ 3).	Chronic diseases that may cause an increase in oxidative stress levels (cardiovascular diseases, autoimmune diseases, cancer, hepatic and renal failure, inflammatory diseases, type 2 diabetes mellitus, obesity, cancer, or atherosclerosis)
Patients who have not received treatment (topical, conventional systemic, or biological targeted treatment) for at least 3 months before enrollment	Other chronic diseases (except psoriasis) that may cause an alteration of the gut microbiota (inflammatory bowel diseases, type 2 diabetes mellitus, obesity, or cardiovascular diseases)
Patients older than 18 years old	Patients younger than 18 years old
Patients who have not received systemic antibiotic therapy for at least 6 months prior to enrollment	Pregnant patients
Patients who agreed to take part in the study and signed the informed consent form	Chronic psychiatric diseases that prevent the patient from understanding the purpose of the study and signing the ethical agreement form

## Data Availability

Data supporting the reported results can be obtained by emailing codrutadobrica@gmail.com.

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
