# Peer review of "Correlations between Gut Microbiota and Hematological, Inflammatory, Biochemical and Oxidative Stress Parameters in Treatment-Naïve Psoriasis Patients"

_ijms, 2024, doi:10.3390/ijms25126649_

Round 1

Reviewer 1 Report

Comments and Suggestions for Authors

This is a significant study on an important issue: the correlations between gut microbiota and hematological, inflammatory, biochemical, and oxidative stress parameters in psoriasis patients

The authors claimed that even though the number of included patients is small, results may be a starting point for future research on the involvement of microbiota-inflammation-oxidative stress axis in psoriasis development.

The purpose is well identified as well as the study limitations.

The work is well outlined, well organized, and relevant. The study is well organized and the results support the discussion. Overall, it is a significant study on this matter. However, in order to be published, in my opinion, there are some major changes to be made:

Major changes:

1)        Page 2, Section Materials and Methods, Line 80-85:  This paragraph does not belong to the Materials and Methods section. Either in the introduction or the discussion.

2)        Page 2, Section Materials and Methods, Line 86-88:  This paragraph should be the beginning of the 2.1. Study group and protocol section.

3)        Page 3, Section Materials and Methods, Line 96, Figure 1:  This is not a Figure. This should be transformed and formatted as a Table within the journal Table format.

4)        Page 3, Section Materials and Methods, Line 100, Figure 2:  This Figure is important but should be in a different section. I would suggest a reorganization in M&M section for example instead of:

2.1. Study group and Protocol

2.2. Ethical aspects

2.3. Assessing oxidant status through Free Oxygen Radical Test (FORT) evaluation

2.4. Assessing antioxidant status through FORD evaluation

2.5. Assessing psoriasis severity and the impact on quality of life of psoriasis patients

2.6. Assessing gut microbiota composition

2.6. Assessing Hematological Parameters and Inflammatory Markers

2.7. Statistical analysis

For example:

2.1. Study group

2.2. Ethical aspects

2.3. Protocol (Figure 2 included here) with a brief description

2.2.1. Oxidant status through Free Oxygen Radical Test (FORT) evaluation

2.2.2. Antioxidant status through FORD evaluation

2.2.3. Psoriasis severity and impact on the psoriasis patients' quality of life

2.2.4. Gut microbiota composition

2.2.5. Hematological Parameters and Inflammatory Markers

2.4. Statistical analysis

5)        Figures 2, 3 and 4 should be modified once they have line numbers included.

In my opinion, this study is important and relevant. It is clearly written and the results support the conclusions. Its limitations are also clearly stated in the document, so after these changes, it can be considered for publication.

Author Response

We acknowledge #Reviewer 1 for the comments that definitely helped us to improve the manuscript

Thank you for the suggestions. We have performed the following changes:

1) We have moved the paragraph in the discussions.

2) We have moved the paragraph in the beginning of Study group section.

3) We have transformed the image into a table and made the changes in the text accordingly.

4) We have reorganized the Materials and Methods section according with the recommendations.

5) We have renumbered the figures accordingly.

Reviewer 2 Report

Comments and Suggestions for Authors

1. Please correct minor editorial errors, such as page 3, second line in the table, is ‘alchool’ should be ‘alcohol’ or page 4 line 140 H2O2 (numbers should be in subscripts).

2. I think figures 3, 4 and 5 are redundant. If the test methods are well described then representing the same in a graphic is unnecessary. I suggest removing this.

3. The parameters shown in Table 1 are for the test group and are compared to normal ranges. Does the ‘normal range’ presented refer to the standard values for these parameters? It may be worthwhile to add values for the control - healthy group in the same table. The indications and descriptions here are not entirely clear and transparent.

4. I would still suggest assessing the level of reduced glutathione as a marker of low-molecular-weight antioxidant compounds.

5. The correlation presented is not enough to talk about interrelationships. I suggest that a chemometric analysis, such as the PCA test, be performed.

Author Response

We acknowledge #Reviewer 2 for the comments that definitely helped us to improve the manuscript.

Thank you for the suggestions. We have performed the following changes:

  1. We have corrected the editorial errors. Thank you for your observation.
  2. We have added additional information in the text and we remove the Figure 3 and 4. Thank you for your observation. We have chosen to keep the Figure 5 (and renumbered it into Figure 2) because we think it is easier and quicker to understand the qPCR processes on the image.
  3. Thank you for your observation. Yes, the normal range represents the standard values for our population. We did not add the values for control because we did not perform a comparative analysis for those parameters between the control group and the psoriasis group. Another study was performed before by our team regarding this aspect, on a larger cohort (Cozma EC, Găman MA, Orzan O, Hamed KV, Voiculescu VM, Găman AM. Oxidative Stress and Inflammation Levels in a Population of Eastern European Naïve Versus Treated Psoriasis Patients. Cureus. 2023 Nov 2;15(11):e48177. doi: 10.7759/cureus.48177.). We wanted on this particular study to focus only on the gut microbiota composition and the correlations between those microorganisms and the parameters evaluated for the psoriasis group.
  4. Thank you for your suggestion. Unfortunately, we only had limited funds for the reagents kits and we decided that for this study was better to focus only on FORD and FORT tests. On the future, we want to extend the study group, to assess the level of reduced gluthatione, 8hydroxi-2-deoxiguanosine and to performe a microbiome sequencing analysis.
  5. Thank you for your response. We have added the analysis.

Round 2

Reviewer 1 Report

Comments and Suggestions for Authors

The manuscript has improved significantly therefore is publishable.